# Predicting 3D forearm fracture angle from biplanar Xray images with rotational bone pose estimation

**Hanxue Gu**[1]               HANXUE.GU@DUKE.EDU
[1] *Department of Electrical and Computer Engineering, Duke, NC, 27703, USA*

**Roy Colglazier**[2]              ROY.COLGLAZIER@DUKE.EDU
[2] *Department of Radiology, Duke, NC, 27703, USA*
**Jikai Zhang**[1]                JIKAI.ZHANG@DUKE.EDU
**Robert Lark** [2]               ROBERT.LARK@DUKE.EDU
**Benjamin Alman**[3]             BEN.ALMAN@DUKE.EDU
[3] *Department of Orthopaedics, Duke, NC, 27703, USA*

**Maciej Mazurowski** [1,2,3]          MACIEJ.MAZUROWSKI@DUKE.EDU
[3] *Department of Computer Science, Duke, NC, 27703, USA*

**Editors:** Accepted for publication at MIDL 2024

## Abstract

Two-dimensional X-ray images, while widely used, have limitations to reflect 3D information of the imaged objects. Several studies have tried to recover such information from multiple X-ray images of the same object. Still, those approaches often fail due to the unrealistic assumption that the target does not move between views and those two views are perfectly orthogonal. A problem where 3D information would be highly valuable but is very difficult to assess from 2D X-ray images is the measurement of the actual 3D fracture angles in the forearm. To address this problem, we propose a deep learning-based method that predicts the rotational movement and skeletal posture from biplanar X-ray images, offering a novel and precise solution. Our strategy comprises the following steps: (1) automatic segmentation of the ulna and radius bones of the forearm on two X-ray images by a neural network; (2) prediction of the rotational parameters of the bones by a pose prediction network; (3) automatic detection of fracture locations and assessment of the fracture angles on 2D images; and (4) reconstruction of the real 3D fracture angle by inferring it from the 2D fracture information and the skeleton pose parameters collected from the two images. Our experiments on X-ray images show that our method can accurately measure 2D fracture angles and infer the pose of the forearm bones. By simulating X-ray images for various types of fractures, we show that our method could provide more accurate measurements of fracture angles in 3D. We are the first attempt for the fully automatic fracture angle measurements on both 2D and 3D versions, and we show the robustness of our method even in extreme cases where the two views are highly nonorthogonal.

**Keywords:** Fracture angle measurement, 3D reconstruction, X-ray measurement

## 1. Introduction

X-ray, known for its low radiation and cost, is the primary choice for diagnosing forearm fractures. However, its 2D projection nature obscures vital details, such as precise fracture angles and posture variations. Specifically, when patients undergo follow-up X-rays to

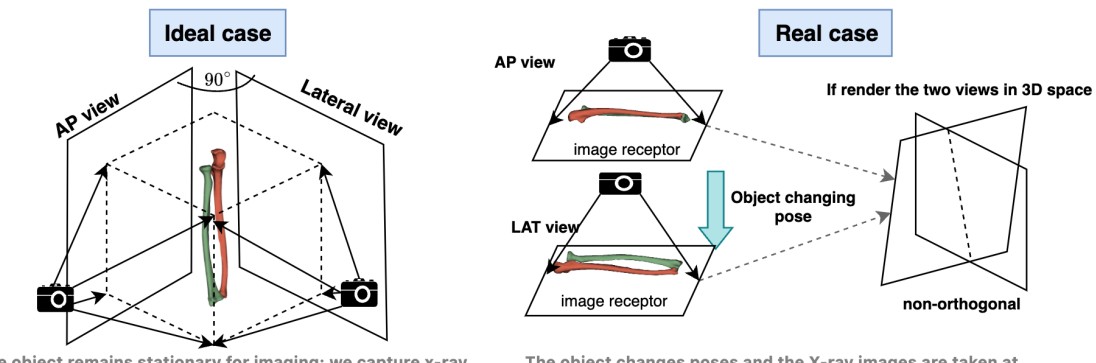

Figure 1: An illustration of a bi-planar view of X-ray images, ideal cases, and real-life clinical practice. The left shows the ideal case where the AP view and Lateral views are orthogonal to each other, while in the real case, the object changes posture across the views and will bring about the misalignment and non-perfect views.

assess forearm fractures, replicating the exact arm positioning from previous images is challenging. This complexity makes it difficult for clinicians to discern whether fracture angulation changes reflect clinical progression or are mere artifacts of different positioning. Such discrepancies can hinder precise diagnosis, surgical preparation, and post-treatment evaluation by physicians (Macken et al., 2022).

Some earlier works (Fotsin et al., 2019; Lamecker et al., 2006; Ehlke et al., 2013; Chen and Fang, 2019; Kasten et al., 2020; Ying et al., 2019; Shrestha et al., 2022; Shiode et al., 2021) attempted to reconstruct 3D bones from biplanar X-ray images. These 3D reconstruction methods using the most commonly applied anteroposterior (AP) view and lateral (LAT) view, rely on the two views being **entirely orthogonal** to one another and the object not changing position or moving during the capture (Gu et al., 2024), see Figure 1 (ideal case). However, in real practice, these two views refer to the anatomical position, which indicates a relatively fixed X-ray shooting angle, and the object's own moving and rotating (see Figure 1 for real cases). Obtaining two views with complete skeletal orthogonality is impossible, especially the limitation of movement caused by fractures; see Appendix A (ill-posed cases) for examples. Therefore, it is usually infeasible to reconstruct the forearm bones from two clinically taken X-ray images with the assumption of orthogonality. A relevant work (Abe et al., 2019) tried to predict the rotational movements of the healthy forearm. However, the required 2D–3D X-ray and CT pairs are difficult to obtain, and their work was limited only to pose estimation.

Meanwhile, fracture assessment is an important task for X-ray-based diagnosis. Some work automatically detects fractures (Guan et al., 2020; Thian et al., 2019) or provides different types of fracture classification (Yadav and Rathor, 2020). Yet, accurately determining fracture angles requires more sophisticated information and has not been accomplished

through a fully automated measurement process. Our algorithm not only achieves the **first** automatic fracture angle measurements but also considers the loss of third-dimensional information caused by X-ray images, which renders forearm posture estimation and reconstructs the fracture angle in 3D.

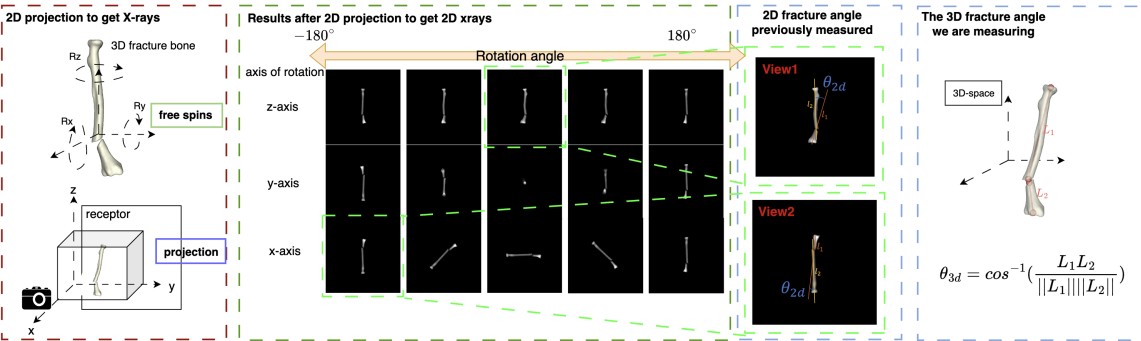

Figure 2: An illustration of the 3D fracture angle we measure. The 3D-coordination system applied in our work was to make the projected receptor in the yz plane. It shows that the rotation around the z-axis and y-axis would influence the 2D angle measured in the 2D X-ray images. The 2D angle varies a lot between different positions of bones, which causes inconsistent angle measurements between X-ray images and possibly inaccurate measurements compared with the real angle.

In our method, we combine (1) the forearm bone segmentation algorithm to extract two forearm bones separately and (2) the pose regression network to predict the posture of bones in two randomly collected views. This process establishes a spatial relationship between the two images in 3D space. (3) a mathematical algorithm to correlate the fracture angles measured in both images with their real-world 3D counterparts. Our method is the first to attempt to achieve fully automatic 2D fracture angle measurements and to reconstruct 3D fracture angles from nonorthogonal 2D X-ray images. It is free from X-ray images and 3D image pairs or planar markers during the training stage, which differs from the previous 2D-3D reconstruction works (Shrestha et al., 2022; Shiode et al., 2021). Our work can also be easily extended to other body parts with 2D X-ray-based fracture angle or bending angle measurements in clinical usage.

## 2. Method

We detail our method for 3D fracture angle prediction in this section. Our algorithm comprised three main components: (1) measure fracture angles in 2D images separately; (2) get the positioning of bones on these two images; (3) combine this information to reconstruct the 3D angulation. The pipeline is illustrated in Figure 3. First, we took two different views of X-ray images from the same patient, segmented the target forearm bones (radius and ulna), and extracted them separately (Section 2.2). Next, we fed the extracted bones to the style-transfer network and the rotational pose estimation network (Section 2.3);

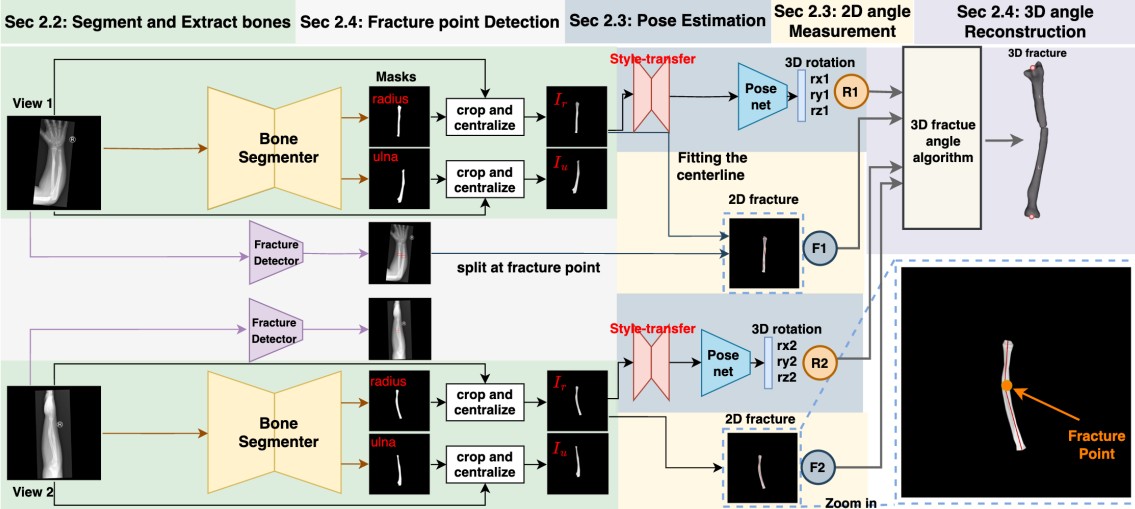

Figure 3: An illustration of the pipeline to get the 3D fracture angle. We represented different steps with different background colors for better illustration. The branch for ulna bones is omitted to save space in this figure.

in parallel, we detected and measured 2D fractures (Section 2.4). Finally, we calculated the 3D fracture angle (Section 2.5).

## 2.1. Datasets Preparation

This retrospective, semi-synthetic study was HIPAA-compliant and approved by the institutional review board (IRB) of Duke University. We collected 1000 forearm X-ray images from the Duke University Health System (DUHS). The X-ray dataset contains 502 AP views and 498 LAT views (*dataset X-ray*). Annotation details are shown in Appendix D. For training the network for pose estimation, we acquired 22 additional CT images and manually segmented the 3D bones (ulna and radius). First, we registered these 3D bones into our defined same standard pose (details are provided in Appendix B), then generated 11,000 digitally reconstructed radiographs (DRR) as simulated X-ray images (dataset *DRR-pose*) by rotating the registered bones around the x-, y-, and z-axes, including $r_x \in [-\pi, \pi]$; $r_y \in [-\pi/6, \pi/6]$; $r_z \in [-\pi, \pi]$ to imitate the real-world situations. Obtaining fracture cases with X-ray and CT image pairs is difficult because such pairs are rarely acquired in clinical practice. To overcome this challenge, we utilized the Blender software (Community, 2018) to simulate and synthesize 40 fracture cases from 20 healthy CT exams, which were used to generate another 4000 DRRs (*DRR-fracture-pose*). Our synthetic fractures contain different conditions with various spins and displacements (shown in Appendix C). More details of the training/evaluation configurations are in Appendix H (Table H).

## 2.2. Segment and extract forearm bones

Considering the necessity to separate two forearm bones with comparable texture, transparency, and potential overlap on an X-ray, we applied instance segmentation, combining detection and segmentation, for separating the two forearm bones. We employed the Mask-scoring R-CNN (Huang et al., 2019), which introduced another mask branch compared with Mask R-CNN (He et al., 2017). When getting the segmented masks $M_r$ and $M_u$ from the input image $I$, we extracted the target bones with $I_r = I * M_r$, $I_u = I * M_u$ and centralized the bones into the middle of the image. The segmentation is trained on *dataset X-ray*, and results and the bone extraction details are shown in Appendix E.

## 2.3. Pose estimation

### 2.3.1. FOREARM BONE POSE ESTIMATION

To train a network for bone pose estimation without manual labels, we utilized the DRR-pose and DRR-fracture-pose datasets. The network, Posenet, adopts a standard Resnet-18 as its core, with the output being a three-dimensional rotation parameter $\mathbf{R} = [r_x, r_y, r_z]$. A $Tanh$ activation followed by a $\pi$ scale is applied post-last layer to normalize the rotation parameters' range. Mean Squared Error (MSE) serves as the training's loss function.

### 2.3.2. STYLE-TRANSFER FOR INFERENCE

Our Posenets, trained on DRRs, encountered a challenge due to the differences between DRRs and real X-ray images, leading to potential drops in performance when applied to actual X-rays. To address this, we implemented a style-transfer technique using Cycle-GAN during the inference phase to bridge the gap between DRRs and X-ray images. This process involved using segmented bone images from X-rays as one domain and DRRs (*DRR-pose* and *DRR-fracture-pose*) as another, enabling the network to adapt X-ray images to the style of DRRs. Further details and results are in Appendix F.

## 2.4. 2D fracture angle estimation

We utilized a Faster-RCNN (Girshick, 2015) network to detect fractures (Guan et al., 2020). When training the fracture detection network, X-ray images (*dataset X-ray*) and DRRs (sampled from *dataset DRR-fracture-pose*) are combined to feed into the network, and the center of the detected fracture region was referred to as the "fracture point" or "breakpoint" to be detected. The measurements of 2D fractures consist of the following three parts: (1) Separate the input image $I_r$ or $I_u$ based on the fracture points; (2) for each view, fit two lines $l_{i1}$ and $l_{i2}$ using centerline extraction and line fitting; and (3) measure the angle $\theta_{2d}$ between the two lines $l_{i1}$ and $l_{i2}$. This section's specifics are detailed in Appendix G.

## 2.5. 3D fracture angle reconstruction

As illustrated in Figure 4, and after previous steps, we got the line segments $l_{11}$ and $l_{12}$ for the image $I_1$, and $l_{21}$ and $l_{22}$ for the image $I_2$, and the rotational pose parameters ($\mathbf{R_1}$ and $\mathbf{R_2}$) for the radius bone in view 1 and view 2, $I_1$ and $I_2$. We calculated the 3D fracture angle $\theta_{3d} = \angle(L_1, L_2)$ using 3D geometry. The real bone centerline segments are

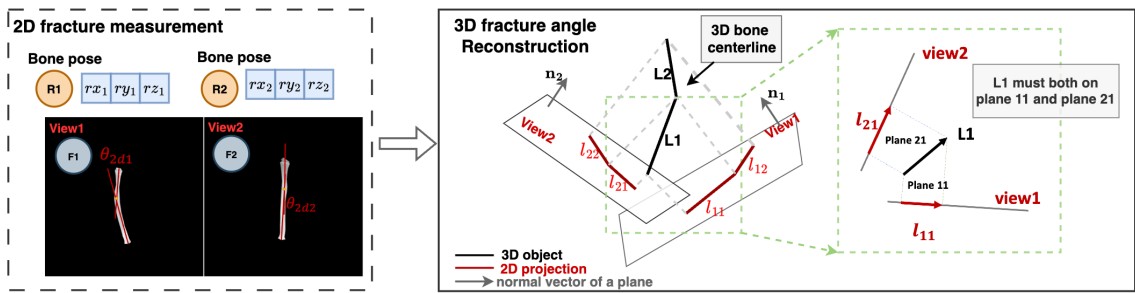

Figure 4: A illustration of the 3D fracture angle algorithm. The algorithm uses the predicted bone postures to reconstruct the 3D bone centerline.

represented in 3D space as $L_1 \in \mathbb{R}^3$ and $L_2 \in \mathbb{R}^3$. The 3D fracture angle was calculated in the following steps. First, we inversed the bone pose matrices $\mathbf{R_1}$ and $\mathbf{R_2}$ to reconstruct the projection planes' direction according to the object and to calculate the normal vectors $\mathbf{n_1} \in \mathbb{R}^3$ and $\mathbf{n_2} \in \mathbb{R}^3$ for *view 1* and *view 2* as the 3D planes $P_1 \in \mathbb{R}^3$ and $P_2 \in \mathbb{R}^3$, see as Figure 4 (right). Next, we highlighted the bone central line segments in 3D space by directional vectors for the 3D lines $\mathbf{l_{11}}$, $\mathbf{l_{12}}$, $\mathbf{l_{21}}$ and $\mathbf{l_{22}}$. Then, we enforced the plane $P_{11} \in \mathbb{R}^3$ through the line $l_{11}$, perpendicular to plane $P_1$, and the plane $P_{21} \in \mathbb{R}^3$ through the line $l_{21}$, perpendicular to plane $P_2$. The normal vectors $\mathbf{n_{11}}$ and $\mathbf{n_{21}}$ can be calculated by: $\mathbf{n_{11}} = \mathbf{L_{11}} \times \mathbf{n_1}; \mathbf{n_{21}} = \mathbf{L_{21}} \times \mathbf{n_2}$.

Because $l_{11}$ is the projection of $L_1$ on plane $P_1$ and $l_{21}$ is the projection of $L_1$ on plane $P_2$, we can infer that $L_1$ is at the intersection of plane $P_{21}$ and plane $P_{11}$, thus the directional vector for $L_1$ can be calculated: $\mathbf{L_1} = \mathbf{n_{11}} \times \mathbf{n_{21}}$; Similarly, we can get $\mathbf{L_2} = \mathbf{n_{12}} \times \mathbf{n_{22}}$. The 3D fracture angle, also known as the angle between three-dimensional lines $L_1$ and $L_2$, can be calculated as follows: $\theta_{3d} = cos^{-1} \frac{\mathbf{L_1} \times \mathbf{L_2}}{\|\mathbf{L_1}\|\|\mathbf{L_2}\|}$.

## 3. Experiments

### 3.1. Bone rotational pose estimation

The bone rotational pose estimation was tested on two datasets. The first dataset was the DRR test set, which consists of 300 simulated X-ray pictures (dataset *DRR-pose-test*). This can be used to evaluate the performance of Posenet under varied rotational parameters. The second dataset includes 10 real X-ray images (dataset *Xray-pose-test*), which also have paired CT images (taken within 3 days). We also included a baseline as a comparison for bone rotational pose estimation (Krönke et al., 2022), which used the segmented bone contours as the input of the pose estimation network.

For the dataset *DRR-pose-test*, since its ground-truth rotational parameters have been known, we evaluated the model's performance by calculating the error between the predicted and ground-truth rotation angles on the x-, y-, and z-axes by the metric of L1-norm among all the available images. We evaluated the performance on the dataset *Xray-pose-test* by visualizing the registered bones (seen in Figure 5). We also analyzed the consistency of the

object shape extracted from the X-ray and the projected DRR under the predicted posture by the shape similarity metric Hu-Moments Contours-match-I1 (Hu, 1962).

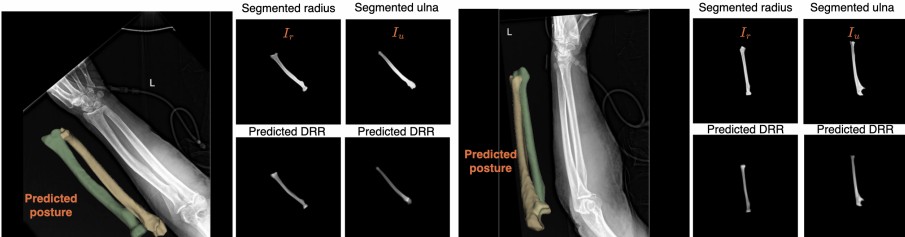

Figure 5: Qualitative results of the 3D bone pose estimation on *xray-pose-test dataset*; The 3D ulna and radius are converted by the estimated 3D transformation matrix and visualized using the 3D slicer software. The predicted DRRs are generated when applying the predicted posture to the CT objects.

Table 1: Quantitative results for the bone pose estimation for *DRR-pose-test* dataset. The rotational error is measured by the mean absolute error (MAE) and standard deviation across 3 repeated training; the 3D rotational angle is defined as the 3D angle to rotate the predicted vector into the ground-truth vector.

| Methods | Bone | rotational error (MAE) at each axis(°) | | | 3D rotational angle(°) |
|---|---|---|---|---|---|
| | | rx | ry | rz | |
| baseline | radius | 4.57±0.36 | 4.12±0.48 | 7.32±0.68 | 9.32±0.41 |
| (Krönke et al., 2022) | ulna | 3.80±0.28 | 2.46±0.32 | 5.85±0.47 | 6.74±0.71 |
| Ours | radius | 2.53±0.31 | 4.26±0.76 | 5.17±0.64 | 6.82±0.96 |
| | ulna | 2.67±0.20 | 3.34±0.15 | 5.33±0.43 | 6.34±0.61 |

### 3.2. 3D fracture estimation

To evaluate our algorithms' effectiveness on 3D fracture angle assessments, we compared our method with two baselines: (1) *Fracture-2D*. This algorithm took only one view of images and measured the 2D fracture angle. This algorithm was similar to (Macken et al., 2022), and detailed in Appendix G; (2) *Fracture-orthogonal*, which ideally assumed that the two input views are orthogonal to each other. The 3D fracture estimation was assessed using two datasets. The first dataset was the simulated DRRs (*3D-fracture-generated-test*), and the ground truth was measured manually by an average of two readers in our institution. Another dataset contains real-life clinical fracture cases, including patients with forearm fractures, noted as (*3D-fracture-real-test*), seen as Table H.

## 4. Results and discussion

The data presented in Table 1 indicate that our pose prediction network can achieve less rotational error than just utilizing the contour information as previous work (Krönke et al., 2022). These results show that for the prediction of the pose, the rotation along the x-axis, which is perpendicular to the projection plane and does not affect the 2D-fracture angle measured during projection, is the easiest to predict. The rotation along the z-axis, as the factor that most affect the fracture angle shown on X-ray images, as depicted in Figure 2, is difficult to predict accurately due to the symmetry of the forearm bones concerning their cylindrical shape (Weinberg et al., 2016). The posture estimation for the ulna exhibits fewer errors. Figure 5 shows the predicted pose of the radius and ulna based on the bones extracted from the input X-ray and the DRR generated from these predicted poses. The average distance between Hu-Moments for the 10 X-ray images is $0.75(\pm0.14)$ for the radius and $0.92\ (\pm0.49)$ for the ulnas.

The average predicted error for our method on the 3D fracture angle prediction is 3.42 degrees with a 95% confidence interval of [3.11, 3.73] for all 600 pairs of images, which is less than the method *Fracture-2D* with an average error of 8.92 and the method *Fracture-orthogonal* with an average error of 5.80. Compared to *Fracture-2D* and *Fracture-orthogonal*, our technique provides more accurate fracture angle assessments ($p < 0.03$). The results of the angle measurements for all the image pairs are shown in Appendix I. (1) relying on a single X-ray to predict fracture angles is inadequate; (2) 2D measurements of fractures vary greatly with the bone's positioning, leading to unreliable outcomes; (3) using dual-view X-rays markedly enhances accuracy; and (4) our method significantly increase measurement precision. This improvement is most notable when the projection angles for the same bone (either radius or ulna) across two images span between 30 and 90 degrees, indicating nonorthogonal viewing angles, as detailed in Appendix I. Our algorithm's measurement of a 3D fracture in a real patient resulted in a 3D fracture angle of 13.01°, which was just 2.06 degrees off from the manual measurements of 10.95°, (seen in Appendix I, Figure 13).

## 5. Conclusion

In summary, our research improves patient care for those with forearm fractures by addressing the challenge of inconsistent arm positioning in two X-rays. This inconsistency can lead to measurement errors and uncertainty for clinicians in distinguishing real changes in fracture angulation from artifacts due to varied positioning. Such uncertainties can complicate determining appropriate treatment strategies, potentially resulting in suboptimal care. Our algorithm uniquely accounts for variations in X-ray positioning, marking a novel approach in 3D reconstruction by considering objects' rotational posture. It enhances diagnostic accuracy, enabling clinicians to more reliably identify true changes in fracture angulation and reduce the risk of recommending unnecessary surgical interventions. We also acknowledged the limitations and discussed future directions, which are discussed in Appendix K.

## Acknowledgments

We would like to express our sincere gratitude to Duke University and the Duke University Health System (DUHS) for their support and resources throughout this project.

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

## Appendix A. Well-posed cases and ill-posed cases

Obtaining two views of forearm X-ray images requires the patient to rotate their elbow and wrist to change their posture, which changes the patient's postural location in both views. As illustrated in Figure 6, the well-posed AP view requires supination of the patient's forearm. Its dorsal surface is to be kept in contact with the cassette while the elbow joint is extended, and the lateral view requires the elbow to be flexed to 90 degrees and the medial aspect of the wrist. The improperly posed examples cannot guarantee positional correction. Therefore, the ulna and radius do not rotate 90 degrees between views.

## Appendix B. Steps for standard pose registration

As depicted in Figure 7, four markers were put on the CT-segmented radius bone for standard pose registration. $F_1$ is set at the end of the styloid process, while $F_2$, and $F_3$ are at two sides; $F4$ is located in the center of the head of the radius. The centroid of this bone is $C$. The standard pose is registered by first moving the bone so that its center aligns with the volume center (which is also the rotation center) and then rotating the bone about the rotational center to ensure that the direction of the vector $< F_4, C >$ correlates with the direction of the z-axis. Finally, we rotate the bone about the z-axis so that the direction of vectors $< F_1, F_2 >$ and $< F_1, F_3 >$ are x-axis symmetric. In Figure 7, the registered bone and the DRR generated by the standard posture are also displayed (upper row, middle box, and right box).

To register the ulna bone, $F_2$ is put at the end of the styloid process of the ulna, and $F_1$ is placed at the head of the ulna. We first moved the centroid of this bone $C$ to match the volume center and then rotated it to make the vector $< F_1, C >$ align with the direction of

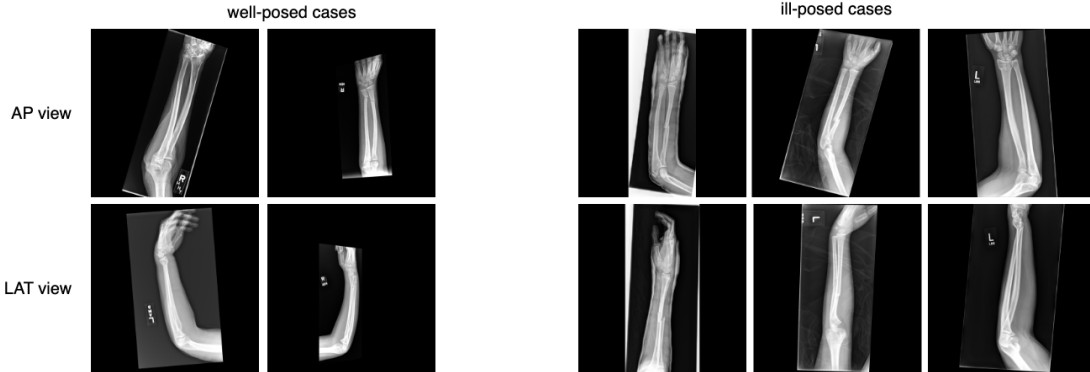

Figure 6: Some examples of the well-posed and ill-posed cases; for the well-posed cases, the ulna and radius bones are nearly orthogonal between AP view and LAT view; and for the ill-posed cases, the rotations of the radius (ulna) between views are hard to predict through human's observation.

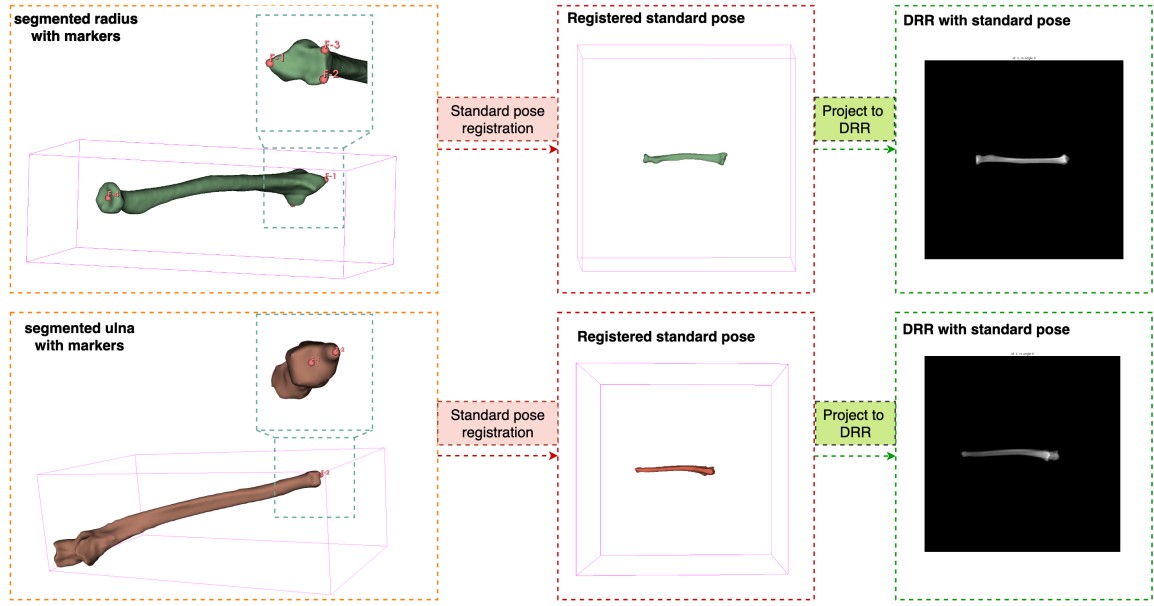

Figure 7: An illustration of the registration pipelines. The upper one is the pipeline for registering a segmented radius into the standard pose, and the lower one is registering a segmented ulna.

the z-axis. Finally, we rotate the bone around the z-axis to make vector $< F_1, F_2 >$ in the direction of the x-axis, seen in the Figure 7 (bottom pipeline).

The markers are set manually in 3D Slicer (Fedorov et al., 2012), and the registration is done automatically through Python. The DRRs for a standard pose correspond to the equation $\mathbf{R} = < rx, ry, rz > = < 0, 0, 0 >$, and the rotation parameters predicted by the pose prediction network (PoseNet) are all relative to this standard pose.

## Appendix C. Example of generated fracture bones

Blender (Beare et al., 2018) is utilized to produce the synthesis fractures. For each bone, we applied a cut at a random location as the fracture point and then misplaced one side of the bone using 3D rotation and translation to simulate the displacement and bending that occurs during bone fracture; examples are shown in Figure 11.

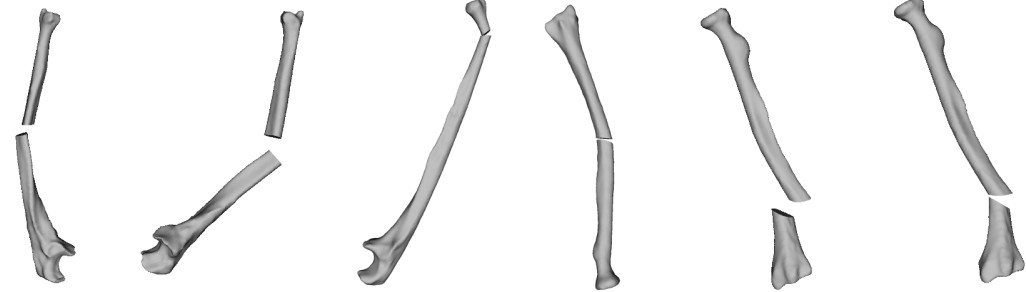

Figure 8: An illustration of the simulated fake fractures, three for ulna and three for radius bones. The bones are broken at the 'fracture point,' and several different deformities with 3D fracture angles exist.

## Appendix D. Details of the annotations

For training segmentation networks, the ulna and radius bones are manually labeled by Labelme (Russell et al., 2008) with the polygon tool. For the fracture detector, at the center of each bone fracture, the fracture location is also labeled for training fracture detection algorithms. If the bones are detached and misaligned, we label each end of the separation with a different marker. The bounding box sizes were set to $50 \times 50$ for all fracture annotations since we only care about the fracture's location in the subsequent steps.

## Appendix E. Segmentation results

The segmented bones are centralized by:

$$\begin{bmatrix} \Delta x_r \\ \Delta y_r \end{bmatrix} = \begin{bmatrix} \frac{h}{2} \\ \frac{w}{2} \end{bmatrix} - \begin{bmatrix} x_{cr} \\ y_{cr} \end{bmatrix}, \begin{bmatrix} \Delta x_u \\ \Delta y_u \end{bmatrix} = \begin{bmatrix} \frac{h}{2} \\ \frac{w}{2} \end{bmatrix} - \begin{bmatrix} x_{cu} \\ y_{cu} \end{bmatrix}$$
$$\mathbf{p_{cr}} = \mathbf{p_r} - \begin{bmatrix} \Delta x_r \\ \Delta y_r \end{bmatrix}, \mathbf{p_{cu}} = \mathbf{p_u} - \begin{bmatrix} \Delta x_u \\ \Delta x_u \end{bmatrix}, \tag{1}$$

where $\mathbf{p_r}$ represents each point in the cropped radius bone $I_r$, and $\mathbf{p_u}$ represents each point in the cropped radius bone $I_u$. $h$ and $w$ are the height and width of the image, respectively. $x_{cr}$ and $y_{cr}$ represent the centroid of the extracted radius ($I_r$), and $x_{cu}$ and $y_{cu}$ represent the centroid of the extracted ulna ($I_r$).

Figure 9 shows some segmentation results and the cropped and extracted examples of bones from the X-ray images. The segmentation model achieves a mean average precision (mAP) (IoU=0.5:0.95) of 0.533 and (mAP) (IoU=0.5) of 0.933 over the evaluation set.

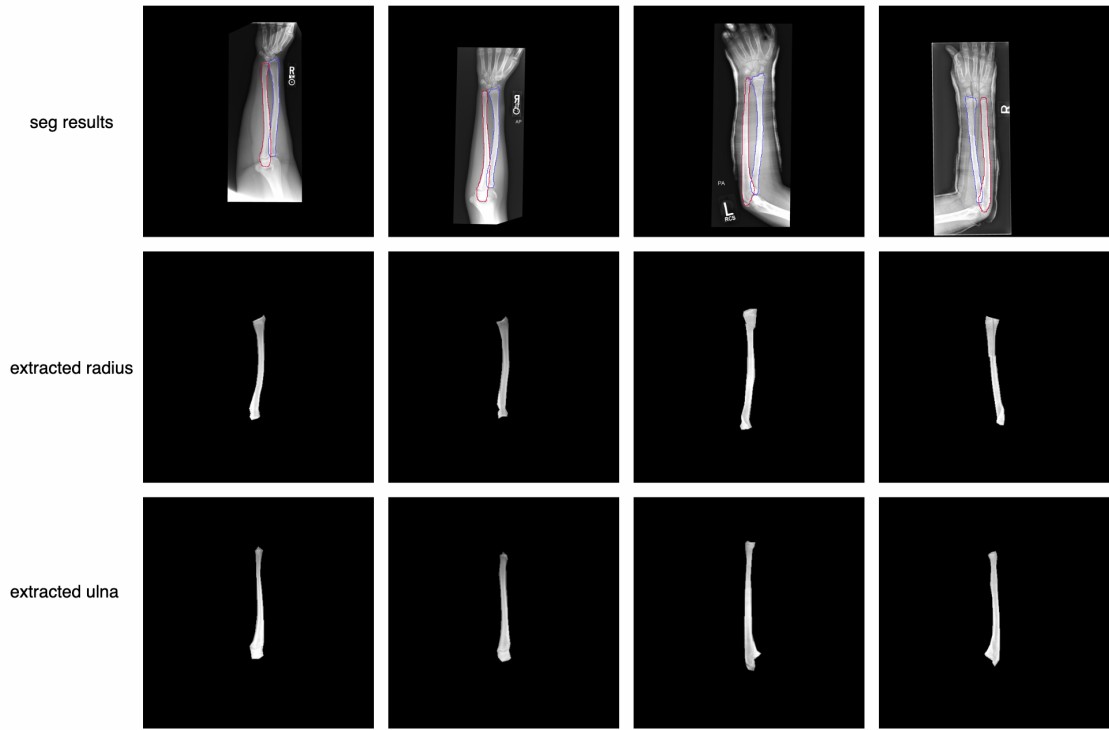

Figure 9: Visualization of the segmentation results for the radius and ulna bones. The first row is the input images with the segmented bone contours, and the second and third rows are the extracted and centralized bones.

## Appendix F. Details of the style-transfer using Cycle-GAN

The Cycle-GAN contains two mapping functions, $G : A \to B$ and $F : B \to A$, and associated adversarial discriminators, $D_B$ and $D_A$. $D_B$ encourages $G$ to translate $A$ into outputs indistinguishable from domain $B$ , and vice versa for $D_A$ and $F$. Two cycle consistency losses are introduced to capture the intuition that if we translate from one domain to another and back again, we should arrive at where we started.

We applied the initial settings of Cycle-GAN, with the extracted bones from X-ray images as domain X, represented by Figure 10 (row 1), and the DRRs as domain Y, defined by Figure 10 (row 3). The network adhered to the original model configuration, adding RandomFlip, RandomRotation, and RandomCropResize to remove the spatial impact of domain information.

We did not use pix2pix (Isola et al., 2017), which was used in (Shiode et al., 2021), for the following reasons: (1) unpaired image translation is required for our situation because 2D and 3D image pairs are difficult to obtain; (2) the cycle setting, which regularizes an inverse pipeline, can ensure the preservation of shape information during style transfer.

The style transfer examples are shown in Figure 10 (row 2) for X-ray images to DRRs and Figure 10 (row 4) for DRRs to X-ray images. During inference of our method, we only employed half of the cycle, which is the $G : A \to B$ for transferring Xray images to DRR style for pose estimation.

## Appendix G. Details of 2D fracture measurements

After we get the 2D extracted forearm bones, $I_r$ or $I_u$, we separate it into two parts by the detected fracture central points. At each part, we applied the Python Fitline library from OpenCV (noa) to get the equations for $l_1$ and $l_1$; then the 2D fracture angle is the angle between these two straight lines.

## Appendix H. Details of the dataset descriptions

The details of the datasets are shown in Table H, including the dataset collection details and the networks' training/validation details. For testing the pose-estimation and 3D-fracture estimation performance, we collected additional test sets, which are ensured to have no overlapping of previous patients. The patient demographic covered in our study is shown in Table 3, where we cover a wide range of patients across different ages or races.

## Appendix I. Visualization of the predicted 3D angles among all the image pairs

Although our method may produce some outliers when the difference between the two views is small (see in Figure 12), i.e., the pose of the bone on the two views varies very little, we believe that these outliers are not a concern considering that in medical practice the views are often provided with an AP view and a LAT view. Even if they are not perfectly perpendicular, image pairs from the AP and LAT views with a bone angle of fewer than 30 degrees are uncommon. The real-patient case is shown in Figure 13, and the

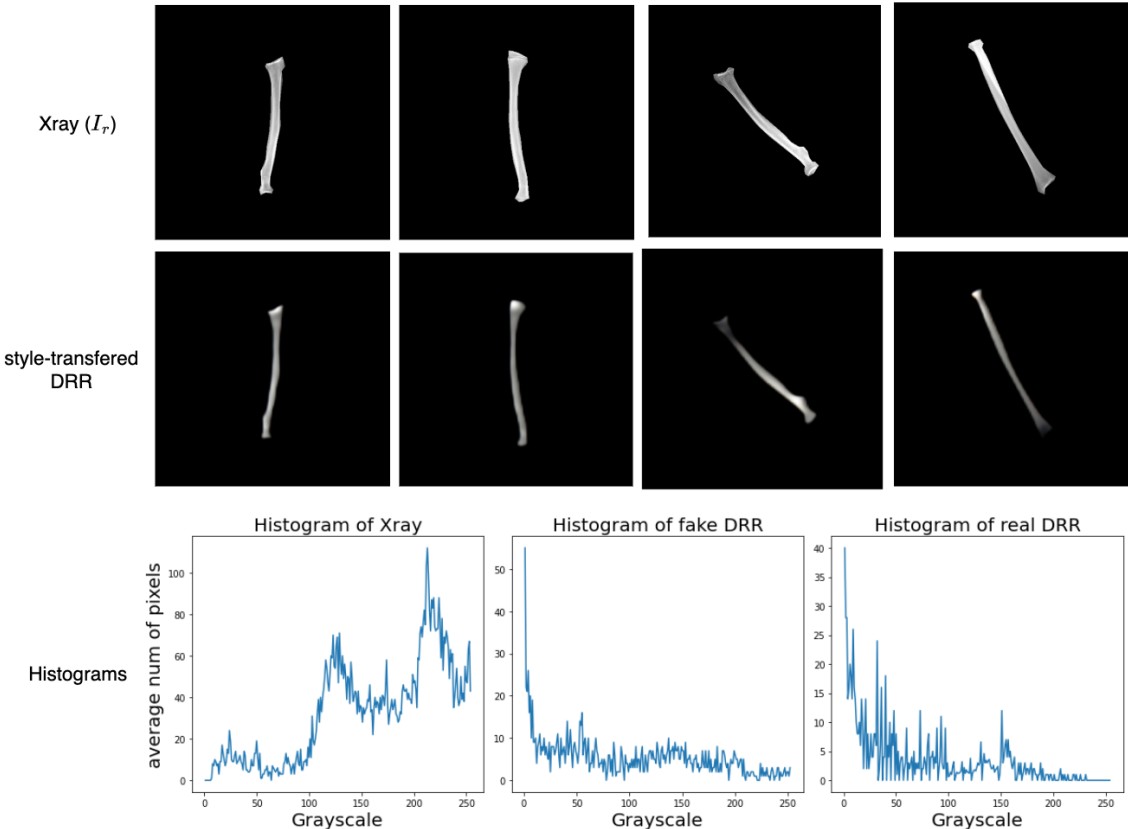

Figure 10: Style-transfer results on both generated DRR and Xray images. The first row is raw X-ray images, and the second is transferred DRRs from X-ray images. The third row is the average histogram of X-ray; our transferred DRR and real-DRRs.

Table 2: The dataset description and the details of dataset split for training/validation.

| datasets | descriptions | stage |
|---|---|---|
| dataset X-ray | 502 AP views and 498 LAT views real xray images | train & validation |
| DRR-pose | 11000 DRRs generated from 22 CTs | train & validation |
| DRR-fracture-pose | generate 40 fracture cases from 20 healthy bones | train & validation |
| DRR-pose-test | 300 simulated X-ray images from 5 untouched CT images | test |
| xray-pose-test | 10 actual X-ray images, which also have corresponding untouched CT image pairs | test |
| 3D-fracture-generated-test | 600 pairs of generated X-ray images of 20 extra simulated fracture instances. The 20 healthy bones are extracted from 10 CTs from 10 new patients | test |
| 3D-fracture-real-test | 1 real-world fracture X-ray and the CT taken on the same day | test |
| **networks** | **dataset used in training stage** | **dataset used in validation stage** |
| segmentation (Mask-scoring R-CNN) | 0.85 * (dataset-Xray) | 0.15 * (dataset-Xray) |
| pose estimation network (Posenet) | 0.85 * (DRR-Pose+DRR-fracture pose) | 0.15 * (DRR-Pose + DRR-fracture bone) |
| style-transfer (Cycle-Gan) | 0.85 * (DRR-Pose+DRR-fracture pose) + 0.85 * (segmented dataset X-ray) | 0.15 * (DRR-Pose+DRR-fracture pose)+0.15*(segmented dataset X-ray) |
| fracture detection network (Faster-RCNN) | 0.85*(dataset-Xray+1000 samples from DRR-fracture-pose) | 0.15 * (dataset-Xray+1000 samples from DRR-fracture-pose) |

| Features | Percentages |
|---|---|
| Sex (male vs. female) | 46.0% |
| Age | |
| - 0-18 | 27.2 % |
| - 19-30 | 13.2 % |
| - 31-40 | 12.0 % |
| - 41-60 | 20.0 % |
| - 61-80 | 17.2 % |
| - 81-100 | 10.4 % |
| First Race | |
| - Caucasian/white | 59.6 % |
| - black | 27.2 % |
| - Asian | 1.6 % |
| - other | 8.8 % |
| - not reported/declined | 2.8 % |

Table 3: Patient Demographic Covered in our Study.

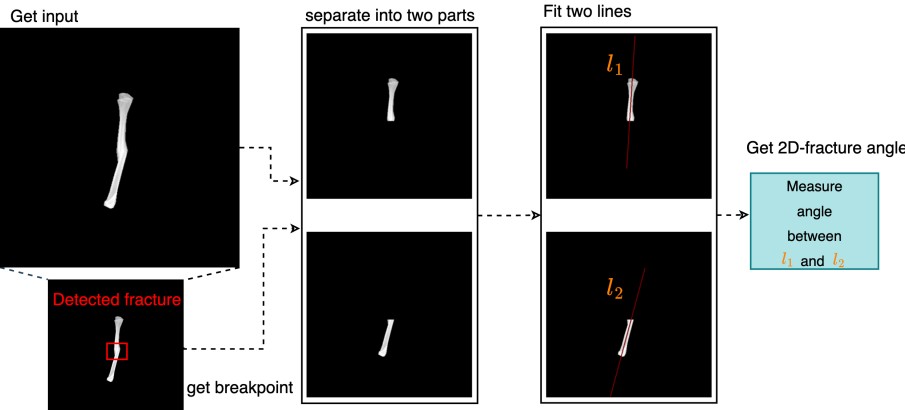

Figure 11: The pipeline for 2D angle measurements. After getting the detected fracture, the bone is separated into two parts, and we do line fitting at each part.

2D measurements of the fracture angle of the ulna bone are 3.54° for view 1 and 12.73° for view 2. The predicted results for method *fracture-orthogonal* is 13.18°.

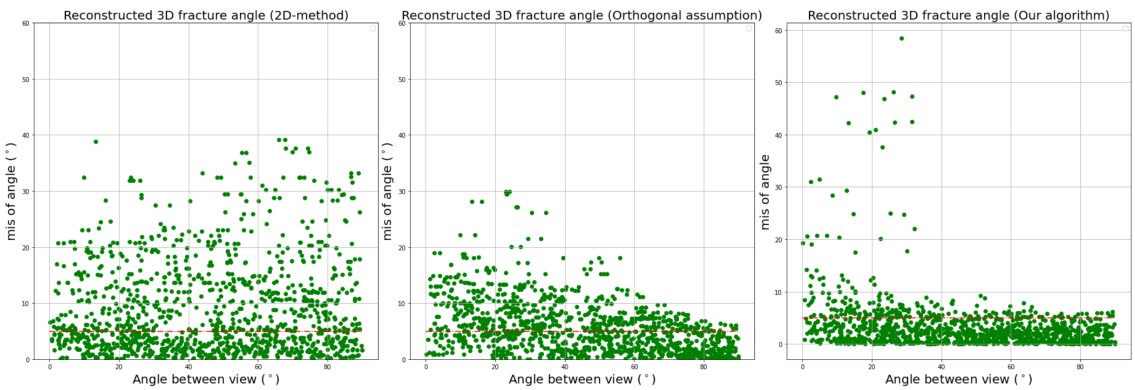

Figure 12: Visualization of the 3D fracture angle prediction performance on dataset *DRR-fracture-generated-test*. From left to right are the (1) prediction on 1-view of the image as a 2D fracture, (2) prediction on 2-view of the image based on an orthogonal assumption, and (3) Our algorithm. For each plot, the x-axis is the angle between two views, and the y-axis is the distance of the predicted angle from the real-measured angle (L1-distance)

.

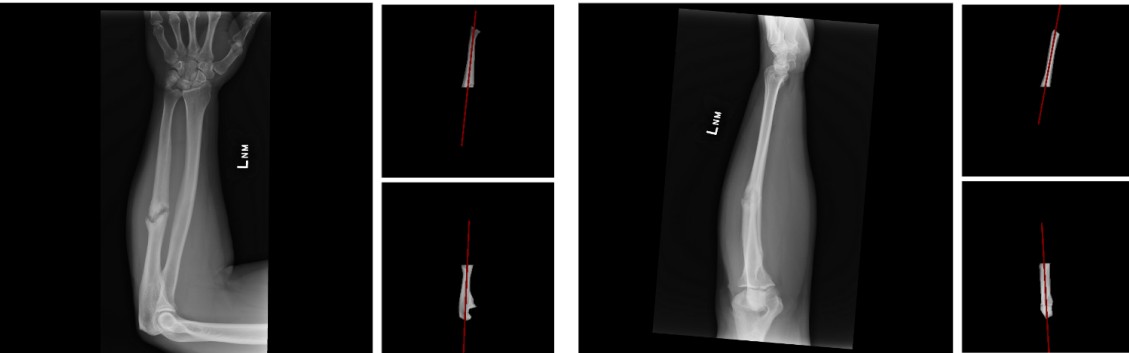

Figure 13: The two view X-ray images of 1 real patient case for fracture measurements. The segmented and extracted ulna bones and fitted lines are on the right side of each image.

## Appendix J. Error Estimation and algorithm robustness analysis

We acknowledge two potential types of noise that might be generated in this study. The first is the "hardware noise" from X-ray imaging, saying incorrect exposure, positioning, and technical competence bring low-quality X-ray images (Waaler and Hofmann, 2010). To evaluate our algorithm's robustness across these "hardware noises," we (1) evaluated the segmentation algorithm across several low-quality images, and as shown in Figure 14, our algorithm could achieve robustness across various qualities of X-rays; and (2) added additional noises into simulated DRRs to evaluate the robustness of the Pose Estimation network. We added Gaussian noise with a $\sigma \sim [0, 0.05]$ distribution. It finally yields 6.46 degrees of error of 3D rotational angle on the ulna bones under a comparable level of clean DRRs.

Another is the "Software noise", which we acknowledge as the main error source from rotational pose estimation, which would influence the norm vector of $n_1$ and $n_2$. Say, we

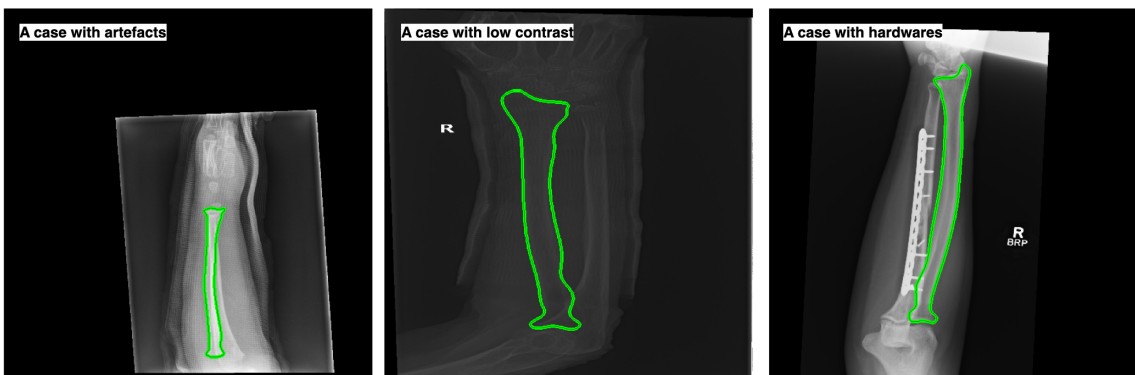

Figure 14: Evaluation of the robustness of bone extraction using various quality Xray images, including (1) images with artifacts; (2) images with low contrasts; (3) images with hardware. We did a postprocessing to smooth the contour for better visualization.

had a perturb $v$ as predicted noise on $n_2$:

$$
\begin{aligned}
n_{21}* &= L_{21} \times (n_2 + v) \\
&= L_{21} \times n_2 + (L_{21} \times v) \\
n_{12}* &= L_{12} \times (n_2 + v) \\
&= L_{12} \times n_2 + (L_{12} \times v) \\
n_{22}* &= L_{22} \times (n_2 + v) \\
&= L_{22} \times n_2 + (L_{22} \times v) \\
L_1* &= n_{11}* \times n_{21}* \\
&= n_{11} \times (L_{21} \times n_2 + (n_1 \times v)) \\
L_2* &= n_{12}* \times n_{22}* \\
&= (L_{12} \times n_2 + (L_{12} \times v)) \times (L_{22} \times n_2 + (n_1 \times v)) \\
&= L_{12} \times n_2 \times L_{22} \times n_2 + L_{12} \times v \times L_{22} \times n_2 + L_{12} \times n_2 \times n_1 \times v + \sigma(\circ).
\end{aligned}
\tag{2}
$$

As our derived 3D angle is calculated, this noise might be accumulated to our final results $\theta_{3d}* = cos^{-1}\frac{\mathbf{L_1}* \times \mathbf{L_2}*}{\|\mathbf{L_1}*\|\|\mathbf{L_2}*\|}$. Based on the empirical results shown in Figure 12, this error is more influential when $\theta = n_1 \times n_2 \to 1$, where AP view and LAT are almost identical to each other. It is quite a rare case in real life.

## Appendix K. Limitations of our work and future directions

We are conscious of the following facts: (1) Initially, the procedure's complexity and detection and segmentation are necessary to measure 2D angles, to which our method adds the measurement of 3D posture and 3D reconstruction. Compared to the end-to-end process, our method does not require many X-ray inputs and 3D output data pairs (which are also essentially unobtainable). It has medically applicable interpretability at each step. (2)

many 3D fracture angles are evaluated based on synthetic fractures with minimal real data. This is because we cannot seek eligible X-ray and CT pairs, and patients do not typically get both modalities simultaneously when fractures occur. We acknowledge this might bring limitations that the types of fractures covered in our study might not be comprehensive, such as the Comminuted Fracture (Rafi and Tiwari, 2023). We found this limitation acceptable because the comminuted fracture would require additional surgical treatment anyway, and the fracture angle measurements would less influence the diagnosis decision. Future expansion of the test set may involve creating phantoms that capture our image pairs. (3) Thirdly, our data was collected mainly on the Duke Health System, which would introduce some bias in the patient population distribution; we would release our codes in the future to get more potential applications and evaluations from different institutions. (4) Lastly, while our 3D fracture measurements are now limited to 3D angle measurements, one of the most important topics for orthopedic practice, we will expand our work to include bone misplacement and even 3D shape reconstruction.

We put our preliminary analysis of measuring bone displacement, as shown in Figure 15. Our current 3D angle estimation shown in Figure 4 mainly considers the angulation relationship between the bone and projection planes, using the plane norm vector. To add the displacement measurement in 3D, as shown in illustration Figure 15 here, we could add the measured distance of two break points in each projection plane ($D_{2D}$), as well as combine the beam to receptor distance $D_s$ and object to source distance ($D_o$) to get the ratio of real distance to distance unit in the projected plane. We have two potential solutions for 3D bone shape reconstruction: one relies on an additional network for bone shape reconstruction, and another solely relies on similar mathematical solutions considering bone thickness when doing the 3D reconstruction (we currently only do centerline).

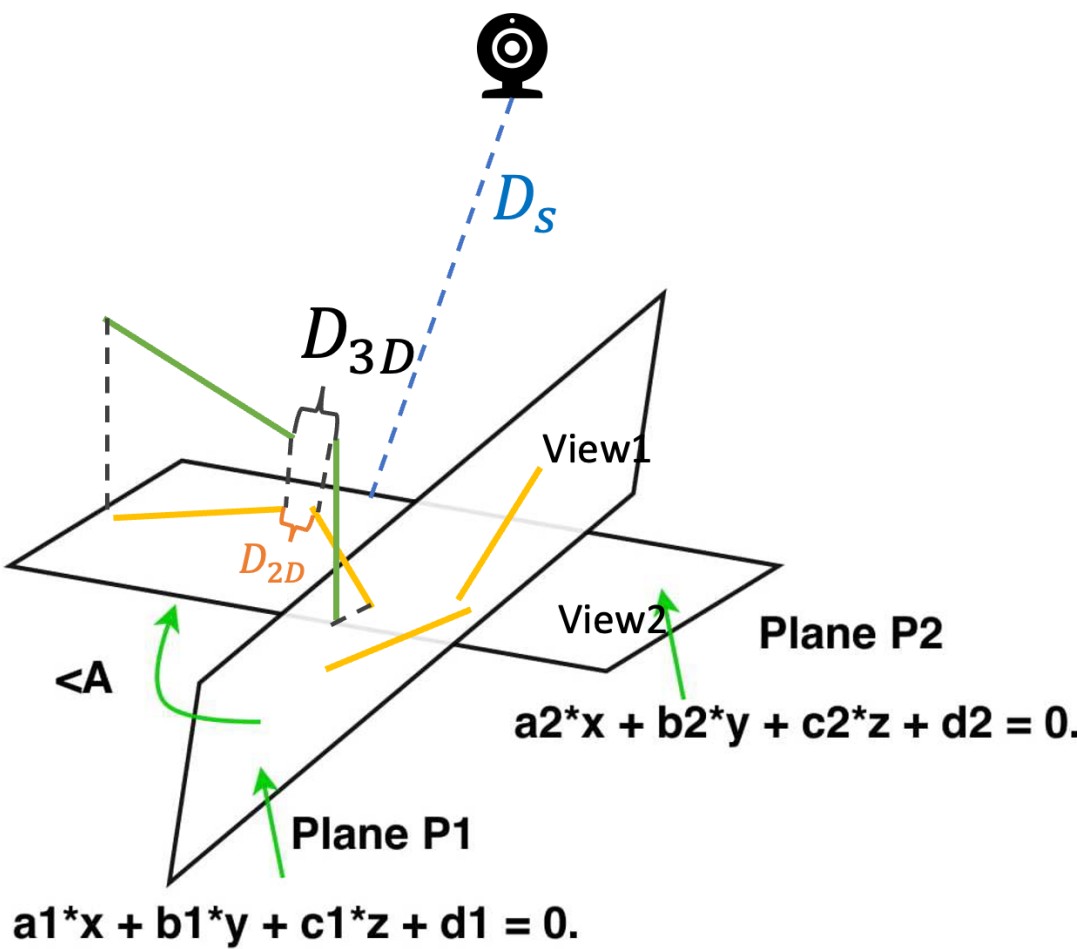

Figure 15: Insights about how to measure the bone displacement in 3D.

