# OpenReview forum: "Predicting 3D forearm fracture angle from biplanar Xray images with rotational bone pose estimation"
_MIDL.io/2024/Conference — MIDL 2024 Poster_

### Official Review · Reviewer_xyYy · 2024-02-21

**Confidence:** 5
**Preliminary Rating:** 1
**Recommendation:** Poster
**Final Rating:** 2.5

**Summary:**

The authors propose a way to compute the angle of a fracture using two non-orthogonal planar radiographs. They collected a dataset comprised of 502 images of AP views and 498 images of LAT views.

**Strengths:**

The idea of computing the 3D angle through a bunch of deep neural network is interesting, starting with the segmentation of the bone in the image. Appendices helped me understand some of the technical parts of the paper.

**Weaknesses:**

The paper reads more like a technical report. The authors mistakenly interchange the terms "X-rays" and "radiographs," with the former referring to the type of electromagnetic radiation that produces the latter.

There are several overly long sentences throughout the article, particularly noticeable in the third sentence of the abstract and the third paragraph of the Introduction.

Many of the figures are too small, making them difficult to interpret. In particular, Figure 3, which is crucial for explaining the proposal, is confusing. The section of the manuscript that references this figure does not adequately justify the choice of methods depicted.

Section 2.1, titled "Dataset Properties," does not appear to accurately present the properties of the dataset; instead, it describes how the authors prepared the datasets for training.

The experiments conducted are somewhat shallow and require improvement. Additionally, Table 1 is presented in an unconventional format.

Minor mistakes:

- A typo in Figure 1: 2D protection -> 2D projection
- Segmentor -> segmenter
- Untouched CT images -> ????

**Detailed Comments:**

See Weakness Section.

**Justification Of Final Rating:**

I appreciate the authors' responsiveness in addressing all of my inquiries. They have indeed done commendable work in rectifying minor issues (although a typo still persists in Fig. 2 with "proection" instead of "projection"). While I still perceive the paper to lean more towards a technical report than a conference paper, the substantial improvement in the writing has influenced my final evaluation, prompting a change in my rating.

**Justification Of The Preliminary Rating:**

The article appears to be in need of further refinement. There are numerous areas for improvement, including the clarity of technical terminology used by the authors and the justification for the selection of each method. Please refer to the "Weaknesses" section for more detailed insights on aspects requiring improvement.

**Questions To Address In The Rebuttal:**

I would like to have a more detailed explanation of the motivations for the proposed framework.
Additionally, how about obtaining approval from the ethics committee?

**Special Issue:**

No

---

> ### Author Response · Authors · 2024-03-17
> **Response to Reviewer xyYy.**
>
> Thanks so much for the reviewer’s advice!
>
> 1. Comments: ''The authors mistakenly interchange the terms "X-rays" and "radiographs," with the former referring to the type of electromagnetic radiation that produces the latter.''
>   - **Answer**: Thanks for pointing out the unclear usage of X-ray and radiographs. We revised all our teams into ‘X-ray images’ which refers to radiography gotten from rays.
>
> 2. Comments: ''There are several overly long sentences throughout the article, particularly noticeable in the third sentence of the abstract and the third paragraph of the introduction.''
>   -  **Answer**: We revised our manuscripts to enhance readability, particularly by shortening the lengthy sentences found.
>
> 3. Comments: '' Many of the figures are too small, making them difficult to interpret. In particular, Figure 3, which is crucial for explaining the proposal, is confusing. The section of the manuscript that references this figure does not adequately justify the choice of methods depicted.''
> -  **Answer**: Thank you for your feedback. We have made updates to the figures in our paper by: (1) enlarging the images and the font sizes displayed within the figures; (2) adding more detailed explanations for the figures both inside figures and captions; and (3) specifically for Figure 3, we have comprehensively revised it by using different colors to delineate various steps and to highlight which section corresponds to this part of the diagram.
>
> 4. Comments: '' Section 2.1, titled "Dataset Properties," does not appear to accurately present the properties of the dataset; instead, it describes how the authors prepared the datasets for training. ''
> - **Answer**: Sorry for the confusion; we amended this section title to 'Dataset Preparation'.
>
> 5. ''Minor mistakes:
> •	A typo in Figure 1: 2D protection -> 2D projection
> •	Segmentor -> segmenter
> •	Untouched CT images -> ????''
>  - **Answer**: We amended all the listed typos here, as well as did another grammar check for the entire manuscript.
>
> 6. ''The experiments conducted are somewhat shallow and require improvement.''
>   - **Answer**: We enhanced our experimental framework by incorporating an evaluation of the algorithm's robustness (see Section J),to expand the scope of our experiments. Please consult our revised paper for more details.
>
> 7. ''I would like to have a more detailed explanation of the motivations for the proposed framework.''
> - **Answer**: We would like to explain the motivations in two aspects, (1) the motivation for solving the clinical task; (2) the motivation for proposing this algorithm and each step's decision making criteria.
>   - For motivation for this clinical task:
>     - When patients undergo follow-up X-ray imaging for a forearm fracture, replicating the exact arm position from previous X-ray images can be challenging. This difficulty can make it hard for clinicians to discern whether observed changes in fracture angulation are genuine or merely artifacts of repositioning. Sometimes, this ambiguity might lead to decisions for surgical intervention due to perceived 'loss of reduction' in the fracture, which could actually result from **different positioning** instead of **true clinical angulation change** [1]. Utilizing our algorithm can significantly minimize these positioning artifacts across X-ray images, potentially sparing patients from unnecessary treatments.
>   - For motivation for this proposed framework.
>     - As we acknowledge that we want to distinguish between **different positioning** and **true clinical angulation change**; we need to accomplish the following tasks in our algoritm (1) measure angles in 2D images; (2) predict the different positioning of bones; (3) combine these informations and reconstruct the real clinical fracture angle in 3D.
>        - (1) measure angles in 2D images, which we chose to segment bones (Section 2.2), and detect where the fracture happens on bone and do 2D measuremnt in (Section 2.4);
>        - (2) predict the different positioning of bones; For this task, we chose to build a Pose prediction network. To accomplish this, we generated DRRs to offer numerous cases without the need for human annotation; then, to make the network trained on DRRs only can also works for X-ray images, we included a Style-tranfer network (Section 2.3).
>        - (3) Reconstruct 3D angle: from which we build our geometrical algorithm using the relationship of two Xray planes.
>        - We added more explanations in Method section about the motivations on each component of algorithm.
>
> 8.'' Additionally, how about obtaining approval from the ethics committee?''
>  - **Answer**: Thanks for the advice, we updated the IRB approval information within our Dataset Preparation section.
>
> 9. ‘’Table 1 is presented in an unconventional format.‘’
>  - **Answer**: Thanks for the advice. We updated table 1's format for better visualization.
>
> Thanks again for your comments and Please let us know if you have further concerns!

---

### Official Review · Reviewer_kcx8 · 2024-02-28

**Confidence:** 5
**Preliminary Rating:** 5
**Recommendation:** Oral
**Final Rating:** 5

**Summary:**

This paper discusses the limitations of traditional 2D X-rays in providing 3D information about imaged objects, particularly in measuring 3D fracture angles in the forearm. The paper proposes a method for reconstructing 3D skeletal structures using bi-planar X-rays, which involves addressing the unrealistic assumption that the target object does not move between the two images and that the two views are perfectly orthogonal. The paper also includes details on the training and evaluation of the network for pose estimation and the segmentation and extraction of forearm bones. The authors plan to expand their work to include misplacement of bones and 3D shape reconstruction.

**Strengths:**

The paper presents a novel approach to reconstructing 3D skeletal structures from 2D X-rays, specifically focusing on the measurement of 3D fracture angles in the forearm. One of the strengths of the paper is its clear and concise presentation of the problem and the proposed solution. The authors provide a detailed explanation of the limitations of traditional bi-planar X-ray-based approaches and how their method overcomes these limitations.

Another strength of the paper is the use of synthetic fractures to evaluate the performance of the proposed method. While the use of synthetic data is not uncommon in computer vision research, the authors provide a clear justification for its use in this particular case, given the difficulty of obtaining eligible X-ray and CT pairs from patients with fractures. The authors also provide a thorough evaluation of their method, including a comparison with state-of-the-art methods and an analysis of the effect of different factors on the performance.

Overall, the paper has strong scientific merits and potential value to the community, as it addresses an important problem in orthopedic practice and presents a promising solution. While the proposed method may not outperform the state of the art on benchmarks, it shows potential for further development and improvement. The structure and language of the paper are appropriate, and the authors adequately address prior work in the field.

**Weaknesses:**

The pape only focuses on the measurement of 3D fracture angles and does not include other important aspects of orthopedic practice, such as the misplacement of bones and 3D shape reconstruction. While the authors acknowledge this limitation and mention their plans for future expansion, it would have been beneficial to include some preliminary results or insights into these areas.

Another weakness of the paper is the reliance on synthetic fractures for evaluation. While the authors provide a clear justification for their use of synthetic data, it is important to note that synthetic data may not fully capture the complexity and variability of real-world fractures. Additionally, the authors mention that they cannot seek eligible X-ray and CT pairs, which raises questions about the generalizability of their method to real-world scenarios.

**Detailed Comments:**

The paper presents a novel approach to reconstructing 3D skeletal structures from 2D X-rays, specifically focusing on the measurement of 3D fracture angles in the forearm. One suggestion for improvement is to include more details on the limitations and assumptions of the proposed method. For example, the authors could discuss the robustness of the method to different sources of noise and error, as well as the generalizability of the method to different view combinations and patient populations. Another suggestion is to provide more context on the clinical relevance of the proposed method.

**Justification Of Final Rating:**

This paper discusses the limitations of traditional 2D X-rays in providing 3D information about imaged objects, particularly in measuring 3D fracture angles in the forearm. The paper proposes a method for reconstructing 3D skeletal structures using bi-planar X-rays, which involves addressing the unrealistic assumption that the target object does not move between the two images and that the two views are perfectly orthogonal. The paper also includes details on the training and evaluation of the network for pose estimation and the segmentation and extraction of forearm bones. The authors plan to expand their work to include misplacement of bones and 3D shape reconstruction.

By supplementing necessary additions and complementing the draft based on the reviewer's comments, this article will be a stronger candidate for an accepted paper. I will sustain my initial evaluation, which was '5: Strong accept.'

**Justification Of The Preliminary Rating:**

With slight weak points of this paper such as focus only on the measurement of 3D fracture angles, this approach is promising and important because 2D images from X-ray modality is common, and thus 3D information obtained from this approach will benefit significantly in actual clinical environments.

**Questions To Address In The Rebuttal:**

I strongly this paper to be accepted with oral presentation.

---

> ### Author Response · Authors · 2024-03-18
> **Response to Reviewer kcx8.**
>
> Dear Reviewer, thanks so much for your comments, and we appreciate you liking our motivation to address real-world clinical challenges. Following the reviewer’s suggestions, we amended these points.
>
> 1. comment: ''The pape only focuses on the measurement of 3D fracture angles and does not include other important aspects of orthopedic practice, such as the misplacement of bones and 3D shape reconstruction. While the authors acknowledge this limitation and mention their plans for future expansion, it would have been beneficial to include some preliminary results or insights into these areas.'' ''One suggestion for improvement is to include more details on the limitations and assumptions of the proposed method. ''
>  - **Answer**:  Following the reviewer’s suggestions, we amended the limitations and assumptions discussion (See updated Appendix K, limitations of our work) on these points.
>      - We included an illustration how to reconstruct 3D displacements by expanding our algorithm.
>      - We added insights and potential solutions to reconstruct shape as well.
>      - We included discussions on other limitations of our study.
>
> 2. comment: ''Another weakness of the paper is the reliance on synthetic fractures for evaluation. While the authors provide a clear justification for their use of synthetic data, it is important to note that synthetic data may not fully capture the complexity and variability of real-world fractures.''
>   - **Answer**: Thanks for pointing out the concerns. We add an additional discussions of the potential limitations about coverage of all real-life fracture types in our limitation section.
>
> 2. comment: ''For example, the authors could discuss the robustness of the method to different sources of noise and error, as well as the generalizability of the method to different view combinations and patient populations. "
>   - **Answer**: We appreciate the reviewer’s suggestion here.
>      - We added an additional error estimation section to discuss the noise sources as well as our algorithm's robustness against these errors, as seen in Appendix J.
>      - We added a demographic recap to our covered datasets and added it to our dataset section Appendix H, as shown in the newly added Table 3, to show the coverage of the population in our study.
>     - We also added additional discussions in the Limitation section to discuss the potential population bias.
> 3. comment: ''Another suggestion is to provide more context on the clinical relevance of the proposed method.''
>   -  **Answer**  Thank you for your recommendation! We have revised our manuscript to incorporate additional clinical motivations for our project into the introduction's first paragraph. Moreover, we have enhanced the conclusion to elaborate on the potential clinical advantages of adopting this method. Please review the updated manuscript for these modifications.
>
>   Thanks again for your comments, and please let us know if you have further questions!

---

> > ### Comment · Reviewer_kcx8 · 2024-03-27
> > **Sustain my initial evaluation**
> >
> > By supplementing necessary additions and complementing the draft based on the reviewer's comments, this article will be a stronger candidate for an accepted paper. I will sustain my initial evaluation, which was '5: Strong accept.'

---

### Official Review · Reviewer_GZFy · 2024-03-02

**Confidence:** 3
**Preliminary Rating:** 4
**Recommendation:** Poster
**Final Rating:** 4

**Summary:**

This paper aims to obtain 3D fracture angle in the forearm from 2 2D bi-planar X-rays under the more realistic assumption that the arm might move during acquisition and the two views are not necessarily perfectly orthogonal. This paper proposes a pipeline for obtaining the 3D fracture angle via (1) segmentation of ulna/radius bone, (2) predict pose parameter of bones in 3D, (3) detect fracture location and measure fracture angle in 2D, and (4) compute 3D fracture angle from 2D fracture angle using pose parameters. The authors demonstrated that this pipeline results in more accurate 3D fracture angle prediction on both synthetic and clinical datasets.

**Strengths:**

+ This paper is solving a challenging problem with realistic assumptions that directly impacts decision making, e.g., doctors use 3D fracture angle for deciding treatments.
+ The models introduced in the pipeline do not rely on paired bi-planar X-ray and CT scans of the same patient for learning. Instead, the authors synthetically generated fractures in CT volumes and rely on DRRs to train these models.
+ This paper is very well written, the main text is easy to follow while the appendix contains detailed information on exactly how a particular step in the pipeline is achieved.

**Weaknesses:**

+ As the authors mentioned in limitations section, this pipeline is complicated with many moving parts, hyperparameters, etc.. It's arguable whether the improved performance justifies the added complexity.

**Detailed Comments:**

+ Grammar/flow of sentences could be improved, in particular in the experiments/results section.

**Justification Of Final Rating:**

I think this paper is good as is and would be a good contribution to this conference.

I also want to comment on Reviewer xyYy (that gives a rating of 1). All weaknesses stated by this reviewer are related to writing & presentation rather than fundamental flaws of the work. I also disagree with the fact that this paper is badly written. In fact, this is one of the most clear paper I've read (best of 6 at MIDL).

Therefore, I'll keep my rating the same, i.e., 4.

**Justification Of The Preliminary Rating:**

This paper solves a real clinically important problem of estimating 3D fracture angle while circumventing the challenge of obtaining paired CT and bi-planar X-rays for learning. The pipeline would be very useful in practice. The paper is well-written. Despite the complexity of the pipeline, I still really liked the fact that this paper is solving a real problem and perhaps the complex pipeline is less of a concern. Therefore, I recommend accepting the paper.

**Questions To Address In The Rebuttal:**

Nothing

**Special Issue:**

No

---

> ### Author Response · Authors · 2024-03-17
> **Replies to the Reviewer's comments.**
>
> Thanks so much for the reviewer’s insightful comments and advice! We addressed the following points:
>
> 1.	Comments: ''Grammar/flow of sentences could be improved, in particular in the experiments/results section''.
>
>  - **Answer**: We have conducted a thorough review and proofreading of our entire manuscript, giving particular attention to the experiments/results section. Furthermore, we have updated the figures within our paper to improve clarity. Please review our revised manuscript.
>
> 2.	Comments: ''As the authors mentioned in the limitations section, this pipeline is complicated with many moving parts, hyperparameters, etc.. It's arguable whether the improved performance justifies the added complexity''.
>
>  - **Answer**: Thanks so much for pointing out the necessity of this algorithmic complexity.
>      - This problem comprises multiple components, and we find that all of which have to be solved to achieve clinically useful results.
>          - Component 1: measure 2D fracture angles, which requires the extraction of bones and the detection of fracture points.
>          - Component 2: Bone posture assessments, while forearm rotation is difficult to evaluate due to the underlying bones being round and a lack of reliable landmarks [1].
>         -  Component 3: 3D fracture reconstruction, from which we build our geometrical algorithm.
>     - For some aspects, we evaluated different choices with simpler solutions to see if they were good enough.
>         -  For example, we compared our intensity-included pose estimation (combine styler-transfer + pose estimation) with a previous contour-based algorithm (shown in Table 1). Our findings, detailed in Table 1, demonstrate superior performance by our method.
>         -  For example, we evaluated our algorithm against a more straightforward reconstruction method based on orthogonal assumptions, which eliminates the need for bone rotational pose estimation or 3D geometrical inversion. As illustrated in the Appendix (Figure 12), this simpler method is over 97.2% more likely to produce larger errors than our algorithm and has a 32.1% likelihood of angulation assessment errors exceeding 10 degrees.
>    - We found this improvement can have more clinical benefits, and the loss of accuracy would result in improper clinical decisions.
>        -  When a fracture occurs, the choice of treatment depends on the degree of angulation and displacement of the fracture and the age of the patient; an angulation of less than 30 degrees and translation of less than 50% is generally accepted, whereas a higher degree of displacement is considered an indication for surgical intervention [2].
>        - During a follow-up examination, clinicians might conclude that surgery is necessary for a patient due to a 'loss of reduction' in the fracture, indicating that the angles observed in the follow-up have not decreased satisfactorily [3].
>        - This means, 10 degrees of error might bring about unnecessary surgical plans both in the fracture occurring stage and the recovery stage.
> Given the complexity of the various components within this clinical problem, along with our supplemental analysis of the clinical advantages provided by this algorithm (which we have highlighted by updating the introduction and discussion sections of our manuscript), we hope the enhanced performance achieved through the algorithm could be justified.
>
> Thanks again for your comments and Please let us know if you have further concerns or advice!
>
> [1] Weinberg DS, Park PJ, Boden KA, Malone KJ, Cooperman DR, Liu RW. Anatomic Investigation of Commonly Used Landmarks for Evaluating Rotation During Forearm Fracture Reduction. J Bone Joint Surg Am. 2016 Jul 6;98(13):1103-12. doi: 10.2106/JBJS.15.00845. PMID: 27385684.
> [2] Macken AA, Eygendaal D, van Bergen CJ. Diagnosis, treatment and complications of radial head and neck fractures in the pediatric patient. World J Orthop. 2022 Mar 18;13(3):238-249. doi: 10.5312/wjo.v13.i3.238. PMID: 35317255; PMCID: PMC8935328.
> [3]	Mehta N, MacFarlane RJ, Brown D. Traumatic disorders of forearm rotation: anatomy, biomechanics and treatment. Br J Hosp Med (Lond). 2014 Feb;75(2):72-7. doi: 10.12968/hmed.2014.75.2.72. PMID: 24521801.

---

### Comment · Area_Chair_kW5X · 2024-03-19
**Paper is open for discussions**

Dear Reviewers The authors have submitted their rebuttal addressing the raised questions. The paper remains open for further discussion and engagement.

---

> ### Author Response · Authors · 2024-03-24
> **Request help from area chair**
>
> Dear Area Chairs:
>
> I hope this message finds you well. I am reaching out to kindly request your assistance regarding the feedback we received from one of our reviewers, identified as reviewer xyYy. This reviewer’s score was “strong reject,” while the other reviewer’s scores were “strong accept” and “weak accept." While we believe that this reviewer is certainly entitled to their opinion that differs from others, the content of their review is not consistent with a “strong reject” score.
>
> Their feedback, primarily focusing on minor aspects such as lengthy sentences, small figures, and terminology differences, does not seem to align with the severity of the score. We are concerned that this might have been an inadvertent error.
>
> Furthermore, the reviewer's comments on our methodological approach were brief, with only half a sentence. This lack of detailed feedback has left us uncertain about how to address these concerns adequately.
>
> Despite these challenges, we have made our efforts to address all the points raised by the reviewer, as well as provide a detailed explanation of the motivation for our work to help understand. However, we have not received any further communication or feedback, which adds to our apprehension about whether our response will be considered.
>
> In this case, we would greatly appreciate your guidance on how to proceed to receive fair consideration of our work.
> Thank you very much for your attention to this matter and your support in navigating this process. We look forward to your advice.
>
> Best regards,
> Authors from submission 277.

---

### Meta-Review · Area_Chair_kW5X · 2024-04-03

**Recommendation:** Accept (Poster)
**Confidence:** 5

**Metareview:**

Two of the three reviewers support accepting this work as a poster and believe it will generate meaningful discussions. I believe the rebuttal has addressed most of the important comments. The authors are requested to revise the camera-ready version in accordance with the feedback provided in the rebuttal.

---

### Decision · Program_Chairs · 2024-04-05

Accept (Poster)